# Psycho-behavioural factors associated with medication adherence among male out-patients with hypertension in a Ghanaian hospital

Irene A. Kretchy[1], Vincent Boima[2]*, Kofi Agyabeng[3], Augustina Koduah[1], Bernard Appiah[4]

1 Department of Pharmacy Practice and Clinical Pharmacy, School of Pharmacy, College of Health Sciences, University of Ghana, Legon, Ghana, 2 Department of Medicine and Therapeutics, University of Ghana Medical School, College of Health Sciences, University of Ghana, Korle-Bu, Ghana, 3 Department of Biostatistics, School of Public Health, College of Health Sciences, University of Ghana, Legon, Ghana, 4 Department of Environmental and Occupational Health, School of Public Health, Texas A&M Health Science Center, TAMU, College Station, Texas, United States of America

* vincentboima@yahoo.com, vboima@ug.edu.gh

**Data Availability Statement:** All relevant data are in the Supporting Information files.

**Funding:** The authors received no specific funding for this work.

## Abstract

Medication adherence is a key health outcome that reflects the health and general well-being of patients with hypertension. Challenges with adherence are common and associated with clinical, behavioural and psychosocial factors. This study sought to provide data on the extent of medication adherence among male patients with hypertension and their biopsychosocial predictors. Patient and clinical characteristics, psychological distress, insomnia and sexual dysfunction were hypothesized to predict outcomes of medication adherence. Utilizing quantitative data from a hospital-based cross-sectional study from 358 male out-patients with hypertension attending a tertiary hospital in Ghana, medication adherence was associated with age, marital status, educational level, income, duration of diagnosis, number of medications taken and sexual dysfunction. These findings support the need for biopsychosocial interventions aiming at promoting adherence while taking these factors into consideration for the benefit of improving the health and general well-being of male patients with hypertension.

## Introduction

Hypertension is an important but treatable public health problem globally. It is estimated to have increased from 442 million in 1990 to 874 million in 2015 [1]. Hypertension is a significant risk factor for cardiovascular disease and stroke: the two leading causes of adult mortality worldwide [1]. Hypertension is also an important public health problem in sub-Saharan Africa. There has been an increase in the prevalence of hypertension in sub-Saharan Africa, and rates in some semi-urban and urban communities are comparable with the prevalence in the United States of America [2, 3]. Prevalence of hypertension in Ghana ranges from 19.2–32.8% in rural areas to 25.5–48% in urban areas [2, 4]. Similarly, studies in Nigeria showed

**Competing interests:** The authors have declared that no competing interests exist.

that prevalence of hypertension in rural areas range from 21 to 25% while in semi-urban and urban areas prevalence ranged from 27 to 46% [5, 6].

Blood pressure control is generally poor among persons with hypertension in sub-Saharan Africa, and efforts to improve blood pressure control are needed [7] There are challenges in the management of hypertension in sub-Saharan Africa in part due to the low rates of hypertension awareness, treatment and control [3]. In Ghana, the reported prevalence of awareness, treatment and control of hypertension were up to 54%, 31% and 13% respectively [2, 4]. For Nigeria, the prevalence of hypertension awareness and treatment were up to 29.4 and 11.3%, while blood pressure control was achieved in 3% of patients with hypertension in community-based studies [3, 6, 8].

The poor blood pressure control among persons with hypertension in sub-Saharan Africa is related to the complex interplay of factors such as lack of knowledge about hypertension, beliefs that are discordant with those of the traditional medical paradigm regarding the causes and treatment of hypertension [9]. Thus, patients' beliefs may be discordant with good practices that help to control high blood pressure [4, 9]. Additionally, patients may not adhere to antihypertensive medications which is the extent to which their medication taking behaviour is consistent with recommendations by their health practitioners because of factors such as the inability to afford the medications [10], co-morbidities including insomnia [11, 12], psychological distress [13] and side effects of the anti-hypertensive medications including sexual dysfunction in men [14–16].

Hypertension and antihypertensive therapy have long been associated with sexual dysfunction (specifically erectile dysfunction) [17]. In 2010, Amidu et al, explored the prevalence of sexual dysfunction among Ghanaian men presenting with various medical conditions and found that the general prevalence of sexual dysfunction was 59.8% of which 50% of this rate was found among men with hypertension [18]. A previous study in South Africa [19] including reviews of studies [20, 21] and guidelines for hypertension management [22] have shown strong associations among sexual dysfunction, hypertension and antihypertensive therapy. Sexual dysfunction is accompanied by psychological problems, emotional stress, somatic complains and social isolation [23, 24]. The relative risk of sexual dysfunction among hypertensive patients is two times higher compared with normotensive patients [25]. Sexual dysfunction among hypertensive patients significantly impact on the quality of life of the patients and their partners [17, 25]. Thus, patients may not adhere to their medications as shown in previous studies [14, 16, 26], resulting in increased risk of morbidity and mortality as a result of complications of hypertension.

To the best of our knowledge, no studies in Ghana have reported sexual dysfunction, psychological distress, and medication adherence of persons with hypertension and their associations to each other given that sexual dysfunction with mixed presentations of mood and behavioural disturbances such as impaired sleep are common in male patients with hypertension and impaired sleep also contributes to the development of psychiatric disorders [27–29], it is necessary to explore this topic in Ghana.

Thus, this study aimed to assess the association among sexual dysfunction, psychological distress, sleep disturbances, and medication adherence in a group of male patients with hypertension in Accra, Ghana.

## Methods

### Study design

This was a single centre, hospital-based cross-sectional study conducted at the Korle Bu Teaching Hospital in the Greater Accra Region of Ghana. Data were collected between January 2017 and April 2017.

## Setting and participants

Three hundred and fifty-eight participants were recruited at the specialist, medical and general outpatient clinics of the hospital. The hospital has 2000-bed capacity with 17 clinical and diagnostic departments. It has an average monthly attendance of 716 patients and an average admission rate of 250 patients per day.

Eligibility criteria for this study included male patients age 18 years and older; diagnosis of hypertension, use of antihypertensive medication for the past 12 months and ability to provide informed consent to participate in the study. Hypertension was defined as Systolic Blood Pressure (SBP) $\geq$ 140mmHg and Diastolic Blood Pressure (DBP) $\geq$ 90mmHg or patients who were on treatment for hypertension. Study questionnaire and other validated data capturing instruments were same for all the specialist units and the general medical outpatient clinics. Participants who satisfied the inclusion criteria were recruited into the study according to the order in which they reported to the out-patient department starting from the first patient. Three research assistants were trained for three days for the interviewer-assisted data collection process. The research assistants read the questions to the respondents, and completed the questionnaires based on the respondents' answers. A minimum required sample size of 351 was obtained using the formula:

$$n_0 = (Deff) \frac{\left(Z_{\frac{\alpha}{2}}\right)^2 p(1-p)}{e^2}$$

Where
$n_0$ = minimum required sample size,
$Z_{\frac{\alpha}{2}}$ = standard normal value of 95% confidence level = 1.96
e = level of precision/margin of error = 0.05
p = prevalence of adherence among patients with hypertension in Ghana and Nigeria = 33.3% [30]
*Deff* = design effect = 1.03.
Assuming a 10% non-response rate, the sample size was computed to be 386.

## Variables

The main study outcome variable was medication adherence. Demographic and other variables such as age, educational background, marital status, average monthly income, presence of comorbidities, number and type of prescribed medications and length of time since diagnosis of disease were assessed. Psycho-behavioural measures of sexual dysfunction, psychological distress, and sleep problems were also recorded.

## Sexual dysfunction

The 15-item International Index for Erectile Function (IIEF) was used to assess sexual dysfunction in the following domains: erectile function, orgasmic function, sexual desire, intercourse satisfaction and overall satisfaction dysfunction [31]. The Erectile Function domain consisted of six questions, Organismic function and Sexual desire–two questions each, Intercourse Satisfaction–three questions, with the Overall Satisfaction having two question. Each of the 15 questions was measured on a scale of 0–5. The questions were measured on a scale of 0–5 with each domain score generated by computing total score for the items in each domain. The scores were negatively scaled implying that lower scores indicated high dysfunction and vice versa [31]. The scale's reliability coefficient in this study based on the Cronbach alpha was 0.959. (Table 1).

**Table 1. The total scores were then categorized into five ordered groups as follows.**

| | Erectile function | Orgasmic function | Sexual desire | Intercourse Satisfaction | Overall satisfaction |
|---|---|---|---|---|---|
| Interpretation (Code used) | Cut-off Score | Cut-off Score | Cut-off Score | Cut-off Score | Cut-off Score |
| Severe dysfunction (5) | 0–6 | 0–2 | 0–2 | 0–3 | 0–2 |
| Moderate dysfunction (4) | 7–12 | 3–4 | 3–4 | 4–6 | 3–4 |
| Mild to moderate dysfunction (3) | 13–18 | 5–6 | 5–6 | 7–9 | 5–6 |
| Mild dysfunction (2) | 19–24 | 7–8 | 7–8 | 10–12 | 7–8 |
| No dysfunction (1) | 25–30 | 9–10 | 9–10 | 13–15 | 9–10 |

## Psychological distress

The 10-item Kessler Psychological Distress Scale measured distress based on the experience of symptoms of anxiety and depression in the most recent 4-week period. The responses to each question was rated from 1 (none of the time) to 5 (all of the time). The total scores ranged from 10 to 50 with scores less than 20, 20–24, 25–29 and 30 and above indicating no, mild, moderate and severe mental disorder respectively [32]. This scale was reliable having a Cronbach alpha value of 0.886.

## Insomnia

Insomnia was measured using the Athens Insomnia scale which assesses eight factors [33]. The first five factors are related to nocturnal sleep and the last three factors identify daytime dysfunction. These factors are rated on a 0–3 scale. The individual's sleep was evaluated from the cumulative score of all factors and reported as their sleep outcome. A cut-off score ≥6 was used to indicate insomnia [33, 34]. The scale's reliability coefficient had a Cronbach alpha of 0.834.

## Medication adherence

The rate of adherence to medications was measured using the Medication Adherence Questionnaire (MAQ) which is quick to administer and score [35]. Each item elicited a 'yes' or 'no' response about patterns of past medication use. High (0), medium (1–2) and low adherence (3–4) were obtained when a patient answered 'no' to all the questions, 'yes' to one question, and 'yes' to two or more questions respectively. The MAQ has shown good validity and reliability in previous studies on cardiovascular disease populations and has also shown good correlation coefficients with other objective measures of adherence and clinical outcomes [35, 36]. In this study the reliability coefficient based on the Cronbach alpha was 0.701.

## Ethics

The study protocol was approved by the Institutional Review Board at the Noguchi Memorial Research Institute for Medical Research, University of Ghana, Legon (035-16/17). A written informed consent was sought from each participant before inclusion in the study.

## Statistical analysis

Data were analyzed using STATA (version 14.1). Descriptive statistics for continuous variables were presented in terms of means and standard deviations for normally distributed data while medians and interquartile ranges were reported for data not normally distributed. Categorical variables were reported in terms of frequencies and percentages. Skewness, kurtosis, and Shapiro-Wilk tests were used to assess the normality of continuous variables. Chi-square test of

independence was used to test for association between the categorical independent variables and medication adherence level. One-way ANOVA test was used to compare the average age across patients' medication adherence levels while the Kruskal Wallis test was used to compare the medians when the data was not normally distributed. Ordered logistic regression models were used to determine associated factors of the level of medication adherence. The likelihood ratio chi-square, Wolfe Gould and Wald tests were used to test for the parallel/proportional odds assumption. The model had level of medication adherence (Low-1, medium-2, and high-3) as the outcome variable with background characteristics, sexual dysfunctions, psychological distress, and insomnia as explanatory variables. All statistical tests were done at 5% significance level. Multiple imputation by chained equation was used to impute for missing information using the predictive mean matching imputation method. Table A shows the questionnaire item response rates (Table A in S1 File).

## Results

### Sociodemographic data

An estimated 2400 patients visited the OPD during the study period and were potentially eligible for the study. Of the1680 patients who were not new cases, 672 were male patients and 386 were approached based on the eligibility criteria. Of the 386 male patients who were eligible and were approached for the study, 358 agreed to participate in the study, representing a 92.7% response rate (Figure A in S1 File). The other 28 participants declined participation on account of fatigue as some of them travel overnight from distant parts of the country to KBTH for review.

The participants had an average age of 56.2±13.50 years (range: 25–91years). The participants were mostly married (70.7%), earned between GH₵ 500 to GH₵ 999 (about US$117 to US$.234) monthly and had either secondary (33.2%) or tertiary (47.8%) education. More than half of the study participants had lived with hypertension for at least five years (52.5%). The participants had a daily average sleep of 7 hours with each person taking a median number of 2 medicines per their medication regiment (Table 2).

### Clinical characteristics of participants

Table 3 presents the levels of sexual dysfunction and other clinical disorders among the study participants. All the study participants (100%) experienced some levels of orgasmic and sexual desire dysfunctions. Dysfunctions between mild and severe levels were also recorded for erectile dysfunction (91.3%), intercourse satisfaction dysfunction (90.2%) and overall satisfaction dysfunction (99.4%).

Insomnia and Psychological distress were prevalent at 63.7% (228/358) and 56.4% (202/358) respectively (Table 3)). The average Psychological distress score was 21.6(range:10–50) and that of medication adherence was 2.1(range:0–4). The categorization of medication adherence scores was low (42.8%), moderate (39.9%) and high (17.9%) (Table 2).

### Factors associated with medication adherence

The bivariate analysis of factors associated with level of medication adherence (Table 4) showed significant associations with age, marital status, educational level, income level, length of diagnosis and number of medications taken ($p < 0.05$). The One-way ANOVA test showed high medication adherence levels among the younger study participants compared with the older ones (low-59.3 ± 13.3 vs medium-55.1 ± 13.2 vs high-51.6 ± 13.0, p<0.001). In addition, all clinical factors except sexual desire dysfunction were significantly associated with the level of medication adherence among the study participants (p<0.05).

**Table 2. Background characteristics of male patients with hypertension receiving treatment at the KBTH.**

|  | Frequency | Percentage |
|---|---|---|
| **Age: Mean ± SD** | 358 | 56.20 ± 13.50 |
| **Marital Status** |  |  |
| Single | 28 | 7.82 |
| Married | 253 | 70.67 |
| Divorced | 56 | 15.64 |
| Widowed | 21 | 5.87 |
| **Educational level** |  |  |
| None | 19 | 5.31 |
| Basic | 49 | 13.69 |
| Secondary | 119 | 33.24 |
| Tertiary | 171 | 47.77 |
| **Income** |  |  |
| Below 500 | 75 | 20.95 |
| 500–999 | 155 | 43.30 |
| 1000–2999 | 103 | 28.77 |
| ≥ 3000 | 25 | 6.98 |
| **Length of Diagnosis** |  |  |
| < 2 Years | 72 | 20.11 |
| 2–4 Years | 98 | 27.37 |
| 5–7 Years | 88 | 24.58 |
| 8–10 Years | 36 | 10.06 |
| > 10 Years | 64 | 17.88 |
| **Number Of Medications: Median (LQ,UQ)** | 358 | 2(2,4) |
| **On non- anti-hypertensive medication** |  |  |
| No | 298 | 83.24 |
| Yes | 60 | 16.76 |
| **Non- anti-hypertensive medication** |  |  |
| **Antidiabetic** | **21** | **5.87** |
| **Statins** | **23** | **6.42** |
| **Antiplatelet** | **26** | **7.26** |
| **Anticoagulant** | **7** | **1.96** |
| **Sleeping Hours: Median (LQ,UQ)** | 358 | 7(6,8) |

SD: Standard deviation, LQ: Lower quartile, UQ: Upper quartile

Testing the parallel or proportional odds assumption using the likelihood ratio chi-square ($\chi^2$ = 4.63, p = 0.705), Wolfe Gould ($\chi^2$ = 39.79, p = 0.390) and the Wald ($\chi^2$ = 40.96, p = 0.342) test confirmed that it was satisfied. From the unadjusted ordered logistic regression models, all the factors were significantly related to the level of medication adherence (p<0.05) except for sexual desire and number of sleeping hours, as shown in Table 4 (S1A). However, the adjusted multiple ordered logistic regression model showed that, age, income level, number of medications, marital status and the number of years the study participants had been diagnosed of hypertension were significantly associated with medication adherence.

The odds of a patient adhering to medication decreases by 3% with every year advancement in age (AOR: 0.97, 95%CI: 0.95–0.99). Study participants who were married had 21% lesser odds of having better medication adherence than study participants who were single.

**Table 3. Distribution of clinical disorders among male patients with hypertension receiving treatment at the KBTH.**

| | Frequency | Percentage |
|---|---|---|
| **Erectal function** | | |
| Mean ± SD | 12.49 ± 8.56 | |
| No dysfunction | 31 | 8.66 |
| Mild dysfunction | 58 | 16.20 |
| Mild to medium dysfunction | 111 | 31.01 |
| Medium dysfunction | 68 | 18.99 |
| Severe dysfunction | 90 | 25.14 |
| **Orgasmic function** | | |
| Mean ± SD | 3.47 ± 2.30 | |
| Mild dysfunction | 25 | 6.98 |
| Mild to medium dysfunction | 114 | 31.84 |
| Medium dysfunction | 116 | 32.40 |
| Severe dysfunction | 103 | 28.77 |
| **Sexual desire** | | |
| Mean ± SD | 4.39 ± 2.11 | |
| Mild dysfunction | 43 | 12.01 |
| Mild to medium dysfunction | 151 | 42.18 |
| Medium dysfunction | 108 | 30.17 |
| Severe dysfunction | 56 | 15.64 |
| **Intercourse Satisfaction** | | |
| Mean ± SD | 6.34 ± 4.39 | |
| No dysfunction | 35 | 9.78 |
| Mild dysfunction | 51 | 14.25 |
| Mild to medium dysfunction | 90 | 25.14 |
| Medium dysfunction | 80 | 22.35 |
| Severe dysfunction | 102 | 28.49 |
| **Overall satisfaction** | | |
| Mean ± SD | 3.75 ± 1.93 | |
| No dysfunction | 2 | 0.56 |
| Mild dysfunction | 22 | 6.15 |
| Mild to medium dysfunction | 116 | 32.40 |
| Medium dysfunction | 117 | 32.68 |
| Severe dysfunction | 101 | 28.21 |
| **Psychological distress** | | |
| Mean ± SD | 21.60 ± 7.75 | |
| No mental disorder | 156 | 43.58 |
| Mild mental disorder | 78 | 21.79 |
| Medium mental disorder | 72 | 20.11 |
| Severe mental disorder | 52 | 14.53 |
| **Insomnia** | | |
| Mean ± SD | 7.64 ± 4.40 | |
| No | 130 | 36.31 |
| Yes | 228 | 63.69 |
| **Medication adherence Level** | | |
| Mean ± SD | 2.07 ± 1.43 | |
| Low | 151 | 42.18 |

(*Continued*)

**Table 3.** (Continued）

|  | Frequency | Percentage |
|---|---|---|
| Medium | 143 | 39.94 |
| *High* | 64 | 17.88 |

SD: Standard deviation

Participants who had been diagnosed for over ten years had 67% less odds of having better adherence compared with those who have been diagnosed for less than two years (Table 5)

## Discussions

Poor adherence to prescribed medications has been reported among patients with hypertension with a number of factors being implicated to play various roles in this health outcomes [26, 37, 38]. To the best of our knowledge, no study has reported the associations among patient characteristics, psycho-behavioural factors and medication adherence in male patients with hypertension in Ghana.

The study observed that medication adherence was affected by age, marital status, educational level, income, duration of diagnosis, number of medications taken and sexual dysfunction.

Socio-demographic factors have been reported to contribute to the medication adherence behaviour of patients with hypertension in general and among hypertensive men in particular [13, 14, 39, 40]. Contrary to other studies where increasing age was associated with improved medication adherence, this study showed that the odds of patients adhering to medication significantly decreased by 3% with every year advancement in age. Previous studies have examined the effect of age on medication adherence with varied results. Some studies have reported high levels of medication adherence with increased age [39, 41] while others have shown otherwise or reported no association between age and medication adherence [11, 42]. For this study participants, as they get older, adhering to their medications become difficult probably due to the increase in the number of medicines taken, increase with years of living with the disease and the experience with sexual dysfunction and other complications either related to the hypertensive disease or as a result of the side effects of the antihypertensive medications. Sexual dysfunction is usually encountered in hypertensive men with hypertension disease and erectile dysfunction increases with age [14]. Similarly, the desire to avoid any challenges with sexual dysfunction may have compelled the married participants to poorly adhere to their medicines compared with their unmarried counterparts.

The findings from this study extend to previous studies on medication adherence where sexual intercourse was perceived as a high priority and patients with hypertension engage in strategies such as discontinuing their antihypertensive medications or selectively adhering to their medications so that they can have sexual intercourse [16, 25]

Lastly, our study revealed that sociodemographic characteristics; income and education were positively associated with medication adherence suggesting that respectively, educated men and men of affluence probably better understood the importance of taking their medication or had better access to their medications with no financial barriers to medications leading to better adherence [40]. Again, this group of participants could communicate their problems with the clinicians for early evaluation and intervention which could help to enhance adherence [36].

**Table 4. Association between background characteristics, clinical disorders and medication adherence level among male patients with hypertension receiving treatment at the KBTH.**

| | Medication Adherence level | | | |
| --- | --- | --- | --- | --- |
| | **Low** | **Medium** | **High** | **p-value** |
| **Age: Mean ± SD** | 59.25 ± 13.34 | 55.06 ± 13.16 | 51.56 ± 12.97 | <0.001 |
| **Marital Status** | | | | 0.001 |
| Single | 11(39.29) | 9(32.14) | 8(28.57) | |
| Married | 121(47.83) | 90(35.57) | 42(16.6) | |
| Divorced | 12(21.43) | 36(64.29) | 8(14.29) | |
| Widowed | 7(33.33) | 8(38.10) | 6(28.57) | |
| **Educational Level** | | | | 0.035 |
| None | 12(63.16) | 5(26.32) | 2(10.53) | |
| Basic | 29(59.18) | 17(34.69) | 3(6.12) | |
| Secondary | 46(38.66) | 48(40.34) | 25(21.01) | |
| Tertiary | 64(37.43) | 73(42.69) | 34(19.88) | |
| **Income** | | | | 0.004 |
| Below 500 | 35(46.67) | 26(34.67) | 14(18.67) | |
| 500–999 | 55(35.48) | 65(41.94) | 35(22.58) | |
| 1000–2999 | 42(40.78) | 48(46.6) | 13(12.62) | |
| 3000–4999 | 19(76.00) | 4(16.00) | 2(8.00) | |
| **Length of Diagnosis** | | | | <0.001 |
| < 2 Years | 27(37.5) | 33(45.83) | 12(16.67) | |
| 2 To 4 Years | 34(34.69) | 42(42.86) | 22(22.45) | |
| 5 To 7 Years | 27(30.68) | 43(48.86) | 18(20.45) | |
| 8 To 10 Years | 16(44.44) | 13(36.11) | 7(19.44) | |
| > 10 Years | 47(73.44) | 12(18.75) | 5(7.81) | |
| **Number Of Medications: Median (LQ,UQ)** | 3(2,4) | 2(1,3) | 2(2,3) | <0.001 |
| **Sleeping Hours: Median (LQ,UQ)** | 7(6,8) | 7(6,8) | 6(5,8) | 0.293 |
| **Clinical Disorders** | | | | |
| **Insomnia** | | | | <0.001 |
| No | 74(56.92) | 37(28.46) | 19(14.62) | |
| Yes | 77(33.77) | 106(46.49) | 45(19.74) | |
| **Erectile Function** | | | | <0.001 |
| No Dysfunction | 20(64.52) | 8(25.81) | 3(9.68) | |
| Mild Dysfunction | 24(41.38) | 23(39.66) | 11(18.97) | |
| Mild to Medium Dysfunction | 35(31.53) | 46(41.44) | 30(27.03) | |
| Medium Dysfunction | 13(19.12) | 40(58.82) | 15(22.06) | |
| Severe Dysfunction | 59(65.56) | 26(28.89) | 5(5.56) | |
| **Orgasimic Function** | | | | <0.001 |
| Mild Dysfunction | 11(44) | 9(36) | 5(20) | |
| Mild to Medium Dysfunction | 49(42.98) | 44(38.6) | 21(18.42) | |
| Medium Dysfunction | 29(25) | 59(50.86) | 28(24.14) | |
| Severe Dysfunction | 62(60.19) | 31(30.1) | 10(9.71) | |
| **Sexual Desire** | | | | 0.338 |
| Mild Dysfunction | 19(44.19) | 18(41.86) | 6(13.95) | |
| Mild to Medium Dysfunction | 59(39.07) | 56(37.09) | 36(23.84) | |
| Medium Dysfunction | 46(42.59) | 47(43.52) | 15(13.89) | |
| Severe Dysfunction | 27(48.21) | 22(39.29) | 7(12.5) | |
| **Intercourse Satisfaction** | | | | <0.001 |

*(Continued)*

**Table 4.** (Continued)

| | Medication Adherence level | | | |
| --- | --- | --- | --- | --- |
| | **Low** | **Medium** | **High** | **p-value** |
| No Dysfunction | 20(57.14) | 11(31.43) | 4(11.43) | |
| Mild Dysfunction | 25(49.02) | 21(41.18) | 5(9.8) | |
| Mild to Medium Dysfunction | 20(22.22) | 43(47.78) | 27(30) | |
| Medium Dysfunction | 29(36.25) | 34(42.5) | 17(21.25) | |
| Severe Dysfunction | 57(55.88) | 34(33.33) | 11(10.78) | |
| **Overall Satisfaction** | | | | 0.013 |
| No Dysfunction | 1(50) | 0(0) | 1(50) | |
| Mild Dysfunction | 7(31.82) | 8(36.36) | 7(31.82) | |
| Mild to Medium Dysfunction | 46(39.66) | 44(37.93) | 26(22.41) | |
| Medium Dysfunction | 41(35.04) | 54(46.15) | 22(18.8) | |
| Severe Dysfunction | 56(55.45) | 37(36.63) | 8(7.92) | |
| **Psychological Distress** | | | | <0.001 |
| No Mental Disorder | 91(58.33) | 43(27.56) | 22(14.1) | |
| Mild Mental Disorder | 25(32.05) | 38(48.72) | 15(19.23) | |
| Medium Mental Disorder | 15(20.83) | 42(58.33) | 15(20.83) | |
| Severe Mental Disorder | 20(38.46) | 20(38.46) | 12(23.08) | |

SD: Standard deviation, LQ: Lower quartile, UQ: Upper quartile.

## Implications for healthcare

Because hypertension is a chronic condition and patients will have to live with it for the rest of their lives, it will be necessary for clinicians to pay attention to older patients and those who have lived with the disease for some time. Although it may be quite a sensitive issue to bring up in patient-healthcare practitioner interactions, clinicians can take the initiative to ask about the sexual health of their patients because having a good sexual function is important for men [43]. Biopsychosocial interventions [44] having pharmacological, psychological and social facets can then be implemented for such patients so that their level of adherence to their prescribed medications will not be compromised for improved quality of life outcomes.

## Implications for policy

As patients live with hypertension in the long term, they are exposed to the long-term effects of the disease and also medications used for treating hypertension. Thus, health practitioners must effectively and efficiently educate their patients regarding hypertension, its treatment and also the implications of non-adherence to their medications. Policy makers should implement measures to make health education on long-term diseases such as hypertension an integral part of medical practice which should be practiced regularly. Also, potential barriers to medication adherence should be included in the national Standard Treatment Guidelines to prompt practitioners to educate patients and advocate for complete adherence to medicines prescribed.

Communication-related interventions including use of the mass media, social media and mobile phones could help reach more adults with effective messages about a need for adhering to antihypertensive medications. Such interventions should also include the benefits of adhering to treatments.

**Table 5. Effects of background factors and clinical disorders on level of medication adherence among male patients with hypertension receiving treatment at the KBTH.**

| | Unadjusted | | | Adjusted | | |
|---|---|---|---|---|---|---|
| | **UOR** | **95% CI** | **p-value** | **AOR** | **95% CI** | **p-value** |
| **Age** | 0.97 | 0.96–0.98 | < 0.001 | 0.97 | 0.95–0.99 | 0.002 |
| **Income** | | | 0.004 | | | 0.044 |
| Below 500 | ref | | | ref | | |
| 500–999 | 1.51 | 0.89–2.55 | | 0.82 | 0.43–1.56 | |
| 1000–2999 | 1.06 | 0.6–1.85 | | 0.51 | 0.25–1.01 | |
| ≥ 3000 | 0.27 | 0.1–0.74 | | 0.24 | 0.07–0.80 | |
| **Sleeping Hours** | 0.98 | 0.89–1.09 | 0.704 | 0.94 | 0.83–1.07 | 0.339 |
| **Educational level** | | | 0.006 | | | 0.186 |
| None | ref | | | ref | | |
| Basic | 1.08 | 0.37–3.16 | | 0.45 | 0.13–1.62 | |
| Secondary | 2.71 | 1.02–7.25 | | 0.87 | 0.28–2.72 | |
| Tertiary | 2.74 | 1.05–7.19 | | 1.10 | 0.36–3.39 | |
| **Number of Medications** | 0.76 | 0.67–0.87 | <0.001 | 0.84 | 0.71–0.99 | 0.035 |
| **Marital Status** | | | 0.035 | | | 0.017 |
| Single | ref | | | ref | | |
| Married | 0.59 | 0.28–1.27 | | 0.79 | 0.32–1.94 | |
| Divorced | 1.14 | 0.49–2.67 | | 1.24 | 0.44–3.48 | |
| Widowed | 1.18 | 0.4–3.48 | | 4.06 | 1.07–15.42 | |
| **Length of Diagnosis** | | | <0.001 | | | 0.011 |
| < 2 Years | ref | | | ref | | |
| 2–4 Years | 1.23 | 0.70–2.16 | | 1.15 | 0.6–2.19 | |
| 5–7 Years | 1.31 | 0.73–2.32 | | 1.57 | 0.8–3.09 | |
| 8–10 Years | 0.87 | 0.41–1.85 | | 1.00 | 0.41–2.43 | |
| > 10 Years | 0.24 | 0.12–0.48 | | 0.33 | 0.13–0.80 | |
| **Insomnia** | | | <0.001 | | | 0.691 |
| No | ref | | | ref | | |
| Yes | 2.25 | 1.47–3.42 | | 1.12 | 0.63–2 | |
| **Erectile Function** | | | <0.001 | | | 0.108 |
| No Dysfunction | ref | | | ref | | |
| Mild Dysfunction | 2.64 | 1.09–6.37 | | 2.41 | 0.75–7.75 | |
| Mild to Medium Dysfunction | 4.25 | 1.88–9.63 | | 2.72 | 0.73–10.11 | |
| Medium Dysfunction | 5.14 | 2.18–12.08 | | 2.29 | 0.56–9.46 | |
| Severe Dysfunction | 0.9 | 0.39–2.1 | | 0.64 | 0.11–3.68 | |
| **Orgasimic Function** | | | <0.001 | | | 0.220 |
| Mild Dysfunction | ref | | | ref | | |
| Mild to Medium Dysfunction | 1 | 0.44–2.28 | | 0.87 | 0.32–2.38 | |
| Medium Dysfunction | 1.85 | 0.81–4.22 | | 1.65 | 0.56–4.84 | |
| Severe Dysfunction | 0.48 | 0.21–1.12 | | 1.55 | 0.41–5.95 | |
| **Sexual Desire** | | | 0.295 | | | 0.967 |
| Mild Dysfunction | ref | | | ref | | |
| Mild to Medium Dysfunction | 1.42 | 0.75–2.68 | | 1.19 | 0.53–2.65 | |
| Medium Dysfunction | 1.05 | 0.54–2.02 | | 1.08 | 0.42–2.78 | |
| Severe Dysfunction | 0.86 | 0.41–1.82 | | 1.2 | 0.41–3.49 | |
| **Intercourse Satisfaction** | | | <0.001 | | | 0.382 |
| No Dysfunction | ref | | | ref | | |

*(Continued)*

**Table 5.** (Continued)

| | Unadjusted | | | Adjusted | | |
|---|---|---|---|---|---|---|
| | **UOR** | **95% CI** | **p-value** | **AOR** | **95% CI** | **p-value** |
| Mild Dysfunction | 1.27 | 0.55–2.94 | | 0.99 | 0.32–3.11 | |
| Mild to Medium Dysfunction | 4.21 | 1.95–9.08 | | 2.05 | 0.63–6.63 | |
| Medium Dysfunction | 2.35 | 1.08–5.12 | | 1.65 | 0.49–5.57 | |
| Severe Dysfunction | 1.03 | 0.48–2.21 | | 2.51 | 0.64–9.79 | |
| **Overall Satisfaction** | | | 0.006 | | | 0.163 |
| No Dysfunction | ref | | | ref | | |
| Mild Dysfunction | 1 | 0.04–24.3 | | 0.15 | 0–4.9 | |
| Mild to Medium Dysfunction | 0.64 | 0.03–14.34 | | 0.06 | 0–1.9 | |
| Medium Dysfunction | 0.67 | 0.03–15.05 | | 0.05 | 0–1.57 | |
| Severe Dysfunction | 0.3 | 0.01–6.68 | | 0.04 | 0–1.52 | |
| **Psychological Distress** | | | <0.001 | | | 0.212 |
| No Mental Disorder | ref | | | ref | | |
| Mild Mental Disorder | 2.49 | 1.48–4.19 | | 1.88 | 1–3.54 | |
| Medium Mental Disorder | 3.41 | 2.01–5.78 | | 1.75 | 0.9–3.4 | |
| Severe Mental Disorder | 2.28 | 1.24–4.19 | | 1.46 | 0.71–3.01 | |

UOR: Unadjusted odd ratio, AOR: Adjusted odds ratio, CI: Confidence interval, ref: reference category

## Strengths of the study

To the best of our knowledge, this is the first study in Ghana to assess medication adherence in male patients with hypertension in order to understand the extent of the challenge for appropriate interventions to be recommended. Again, with a general paucity of information on the psycho-behavioural perspective of medication adherence, the approach we used is a strength of our study.

## Limitations

Our study had some limitations. Although hypertension affects both males and females, this study concentrated only on males to assess male predominant factors associated with the high medication non-adherence rate among male hypertensive patients compared with their female counterparts as reported in previous studies [30]. Also, the perspective of the partners of these male patients were missing. However, with the current evidence from our study, there is a great opportunity for studies involving female patients with hypertension. The use of a cross-sectional design limits the ability to determine the directions of the associations found in this study. Another limitation of this study was the use of self-reported measures for adherence and insomnia instead of objective tools which could affect the right estimation of these levels. In addition, these self-reported measures may be prone to recall bias. To reduce the potential of recall bias, we limited the timeline during which these behaviours occurred to the most recent 4-week period. We also note that, this was a study conducted in a teaching hospital in Ghana so the findings cannot be generalised to all male patients with hypertension in Ghana.

## Conclusion

This study found that the medication adherence behaviour of male patients with hypertension was significantly associated with age, marital status, educational level, income, duration of diagnosis, number of medications taken and sexual dysfunction. Biopsychosocial interventions

aiming at promoting adherence while taking these pharmacological, psychological and social factors into consideration may be beneficial for improving the health and general well-being of male patients with hypertension.

## Supporting information

**S1 File. Supporting Information.** Figure A Table A
(DOCX)

## Acknowledgments

The authors will like to acknowledge the staff and patients at the specialist, medical and general outpatient clinics at the Korle-Bu Teaching Hospital.

## Author Contributions

**Conceptualization:** Irene A. Kretchy, Vincent Boima, Kofi Agyabeng, Augustina Koduah, Bernard Appiah.

**Data curation:** Irene A. Kretchy, Bernard Appiah.

**Formal analysis:** Irene A. Kretchy, Kofi Agyabeng.

**Investigation:** Irene A. Kretchy, Vincent Boima, Kofi Agyabeng, Augustina Koduah, Bernard Appiah.

**Methodology:** Irene A. Kretchy, Vincent Boima, Kofi Agyabeng, Augustina Koduah, Bernard Appiah.

**Project administration:** Irene A. Kretchy, Bernard Appiah.

**Supervision:** Irene A. Kretchy, Vincent Boima, Augustina Koduah, Bernard Appiah.

**Validation:** Irene A. Kretchy, Vincent Boima, Kofi Agyabeng.

**Visualization:** Irene A. Kretchy, Vincent Boima, Kofi Agyabeng, Augustina Koduah.

**Writing – original draft:** Irene A. Kretchy, Vincent Boima.

**Writing – review & editing:** Irene A. Kretchy, Vincent Boima, Kofi Agyabeng, Augustina Koduah, Bernard Appiah.

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
