## [Decision Letter · Decision Letter 0]

31 Oct 2019

PONE-D-19-23246

Psycho-behavioural factors associated with medication adherence among male out-patients with hypertension in a Ghanaian hospital

PLOS ONE

Dear Dr. Boima ,

Thank you for submitting your manuscript to PLOS ONE. After careful consideration, we feel that it has merit but does not fully meet PLOS ONE’s publication criteria as it currently stands. Therefore, we invite you to submit a revised version of the manuscript that addresses the points raised during the review process.

We would appreciate receiving your revised manuscript by 30. November 2019. To enhance the reproducibility of your results, we recommend that if applicable you deposit your laboratory protocols in protocols.io, where a protocol can be assigned its own identifier (DOI) such that it can be cited independently in the future. For instructions see: http://journals.plos.org/plosone/s/submission-guidelines#loc-laboratory-protocols

We look forward to receiving your revised manuscript.

Kind regards,

Tim Mathes

Academic Editor

PLOS ONE

Journal Requirements:

3. Please state in your methods section the participant recruitment date.

4. Please provide additional details regarding participant consent. In the ethics statement in the Methods and online submission information, please ensure that you have specified what type of informed consent you obtained (for instance, written or verbal). If consent was verbal, please specify how you recorded/documented participant consent and whether your ethics committee approved this consent procedure.

Additional Editor Comments:

This is an interesting manuscript. However, the reviewers and I have some major concerns that should be revised regarding the methods before I it can be published.

In addition to the comments of the reviewer please consider the following issues.

Methods

- Please report the results according the STROBE guideline as far as possible

- Please provide all information on the assumptions of the sample size calculation

- Please provide a definition for adherence

- Please specify how adherence was categorized for the logistic regression analysis

- Please specify how continuous predictors were categorized for the logistic regression

- Please clarify why you perform two univariate analysis that answer the same question (table 3 and table 4). Was one of the analyses planned as sensitivity analysis? In particular, the Chi-square analysis in table 3 and univariate analysis of categorical variables in table 4 is redundant, which might be confusing for the reader.

Results

- Please describe the patient-flow in detail (e.g. using a flow-chart). In addition, information on missing data for the outcome as well as predictors should be provided.

- You does not perform a confirmatory study with an a-priory defined hypothesis, Therefore, please delete all signs (*) to indicate statistical significance below the tables

- The value of the Chi-square statistic can be deleted

- If written in English, please provide the study protocol as supplemental material.

- Please indicate if the study was registered in any trials registry

Reviewers' comments:

Reviewer's Responses to Questions

**Comments to the Author**

1. Is the manuscript technically sound, and do the data support the conclusions?

Reviewer #1: Partly

Reviewer #2: Yes

Reviewer #3: Yes

2. Has the statistical analysis been performed appropriately and rigorously? 

Reviewer #1: I Don't Know

Reviewer #2: Yes

Reviewer #3: No

3. Have the authors made all data underlying the findings in their manuscript fully available?

Reviewer #1: Yes

Reviewer #2: Yes

Reviewer #3: Yes

4. Is the manuscript presented in an intelligible fashion and written in standard English?

Reviewer #1: No

Reviewer #2: Yes

Reviewer #3: Yes

5. Review Comments to the Author

Reviewer #1: Thank you very much for giving an opportunity to review the present manuscript. The authors have evaluated the psycho-behavioural factors associated with medication adherence among male patients with hypertension in Ghana. They found that medication adherence of male hypertensive patients was significantly associated with age, marital status, educational level, income, duration of diagnosis, number of medication taken, and sexual dysfunction. This study is quite interesting in point that it evaluates medication adherence from the perspective of behavioural and psychosocial, as well as clinical factors. I think however that there are some improvements that should be made before publication. And the number of pages and lines should be described in the manuscript, because it is hard to point out.

[Methods]

Participants (or limitation section)

1. As it is stated in the “limitations”, more detailed reason is necessary why this study population was only male patients.

Measures

2-1. Is the “type of prescribed medication” only antihypertensive agent? And, type of medication (e.g. antidiabetic agent) is not seen in Table 1. If you investigate the type of prescribed medication other than antihypertensive agent, it should be added in the results section and Table 1.

2-2. Did you analyse the “patients comorbidity” as clinical characteristics? I think that comorbidity is one of the most important factors that affect medication adherence in patients with chronic diseases.

[Results]

Background and clinical characteristics

3-1. An average age of 56.2±SD?

3-2. Erectile dysfunction (92.3%)? It is 91.3% in Table 2.

[Discussions]

4-1. Socio-demographic factors have been reported to contribute to the medication adherence behaviour of patients with hypertension…

References are required in this sentence.

4-2. Similar to other studies where increasing age was associated with improved medication adherence, this study showed that the odds of patients adhering to medication significantly decreased by 3% with every year advancement in age.

I think “Similar to” is incorrect. The results of this study showed that increasing age was associated with “poor” medication adherence.

4-3. For this study participants, as they get older, adhering to their medications become difficult probably due to the increase in the number of medicines taken, increase with years of living with the disease and the experience with sexual dysfunction and other complications either related to the hypertensive disease or as a result of the side effects of the antihypertensive medications…

You should analyse the relationship between age (e.g. younger (<65) vs older (>65)) and the number of medicines taken, the length of diagnosis, sexual dysfunction and other complications, and discuss about those comparing with previous reports.

4-4. Lastly, among the sociodemographic characteristics, income and education were associated with medication adherence...

It is unclear which part is derived from the data of this study or that of previous reports. You should re-organize this part.

4-5. Implications for healthcare

You should explain “biopsychosocial interventions” in detail to the readers to understand using previous reports.

4-6. Implications for healthcare

Although it may be quite a sensitive issue to bring up “during” patient-healthcare practitioner interactions...

during?

4-7. Implications for policy

practised → practiced?

Reviewer #2: In this study, authors set out to investigate medication adherence among male patients with hypertension, as well as factors associated with adherence with a focus on psycho-social determinants including sexual dysfunction and sleep difficulties. A number of factors were identified, and authors suggest the potential contribution of these psycho-social factors in medication (non)adherence. The paper is generally well written. I have few comments/questions for your consideration.

1) How did you arrive at this sample size? There is limited information on the sampling strategy. Simply stating participants were randomly recruited seems insufficient. How was the sampling randomization done? In addition, how many potentially eligible participants were approached/invited prior to obtaining final study sample? Response rate?

2) A number of measurement tools/questionnaires were used to assess sexual dysfunction, insomnia and medication adherence. Have they been previously validated in similar Ghanaian populations? Otherwise, it might be good to comment on how you assured validity of these tools in your study.

3)Almost half of your study population had tertiary education. This seems quite high. Is this representative of the Ghanaian population? Was there some form of selection bias author may want to comment on?

4) At the beginning of your results section, you mention the average age, and after you write 'SD'. Can you please provide the actual standard deviation.

5) This study is from a single-centre and hospital based. In the limitations, please, provide further discussion on the external validity of your study findings.

Secondly, you want to consider discussing further the limitations of self-reported tools (and compared to objective measures) as used to assess adherence, insomnia, etc.

Thank you.

Reviewer #3: 1. Although authors try to mention under limitations section why conducted only on male, still need further clarification

2. Why authors not used the standard tool for assessing adherence, e.g MMAS-8? Need to mention and describe it.

3. Analysis section has problems, e.g Multivariate analyses should include the percentage/frequency's of each variable with respec to adherent vs non adherent. Require extensive revision in this part.

6. PLOS authors have the option to publish the peer review history of their article (what does this mean?). If published, this will include your full peer review and any attached files.

Reviewer #1: Yes: Motoyasu Miyazaki

Reviewer #2: No

Reviewer #3: No

---

## [Author Response · Author response to Decision Letter 0]

18 Nov 2019

Response to Reviewers has been attached as part of this submission 

Thank you

---

## [Decision Letter · Decision Letter 1]

6 Dec 2019

PONE-D-19-23246R1

Psycho-behavioural factors associated with medication adherence among male out-patients with hypertension in a Ghanaian hospital

PLOS ONE

Dear Dr. Boima

Thank you for submitting your manuscript to PLOS ONE. After careful consideration, we feel that it has merit but does not fully meet PLOS ONE’s publication criteria as it currently stands. Therefore, we invite you to submit a revised version of the manuscript that addresses the points raised during the review process.

A requirement for acceptance is that the results are reported according the STROBE statement for cross-sectional studies: https://www.strobe-statement.org/index.php?id=available-checklists 

In particular, information on patient flow/missing data and (avoiding) potential bias  should be provided. 

We would appreciate receiving your revised manuscript by 14.12.2019. To enhance the reproducibility of your results, we recommend that if applicable you deposit your laboratory protocols in protocols.io, where a protocol can be assigned its own identifier (DOI) such that it can be cited independently in the future. For instructions see: http://journals.plos.org/plosone/s/submission-guidelines#loc-laboratory-protocols

We look forward to receiving your revised manuscript.

Kind regards,

Tim Mathes

Academic Editor

PLOS ONE

Reviewers' comments:

Reviewer's Responses to Questions

**Comments to the Author**

1. If the authors have adequately addressed your comments raised in a previous round of review and you feel that this manuscript is now acceptable for publication, you may indicate that here to bypass the “Comments to the Author” section, enter your conflict of interest statement in the “Confidential to Editor” section, and submit your "Accept" recommendation.

Reviewer #1: All comments have been addressed

Reviewer #2: All comments have been addressed

2. Is the manuscript technically sound, and do the data support the conclusions?

Reviewer #1: Yes

Reviewer #2: Yes

3. Has the statistical analysis been performed appropriately and rigorously? 

Reviewer #1: Yes

Reviewer #2: Yes

4. Have the authors made all data underlying the findings in their manuscript fully available?

Reviewer #1: Yes

Reviewer #2: No

5. Is the manuscript presented in an intelligible fashion and written in standard English?

Reviewer #1: Yes

Reviewer #2: Yes

6. Review Comments to the Author

Reviewer #1: (No Response)

Reviewer #2: Authors of this manuscript have addressed most of my comments/concerns. I have no further major comments. Thank you.

7. PLOS authors have the option to publish the peer review history of their article (what does this mean?). If published, this will include your full peer review and any attached files.

Reviewer #1: Yes: Motoyasu Miyazaki

Reviewer #2: No

---

## [Author Response · Author response to Decision Letter 1]

16 Dec 2019

I am submitting a revised manuscript entitled “Psycho-behavioural factors associated with medication adherence and health-related quality of life of male out-patients with hypertension in a Ghanaian hospital” for publication in PLOS ONE journal. All comments have been addressed point-by-point as suggested by the reviewers. There are no legal restrictions on sharing a de-identified data set. The results have been reviewed according to STROBE statement for cross-sectional studies. 

All authors declare no conflict of interest. 

I will be grateful if this manuscript can be considered for publication in your journal. 

Thank you. 

Yours Sincerely

Dr. Vincent Boima

---

## [Editor Report · Decision Letter 2]

19 Dec 2019

PONE-D-19-23246R2

Psycho-behavioural factors associated with medication adherence among male out-patients with hypertension in a Ghanaian hospital

PLOS ONE

Dear Dr. Boima ,

Thank you for submitting your manuscript to PLOS ONE. After careful consideration, we feel that it has merit but does not fully meet PLOS ONE’s publication criteria as it currently stands. Therefore, we invite you to submit a revised version of the manuscript that addresses the points raised during the review process.

There is still no information on missing data in the statistic section (STROBE item 12c) as well as in the results section (STROBE item 13).

I cannot imagine that all patients who agreed to participate provided fully complete questionnaires (i.e. no missing answer at all). Do you include only participants with complete questionnaires (i.e. without any missing response to any variable)? This means you performed a complete case analysis. If so, please describe this in the publication. 

Otherwise pleas specific how you handled missing responses (e.g. mean imputation) and in the case information on the adherence measures (outcome) in addition information on the amount of missing values.  

In addition, please provide information who performed the assessment. Were the patients interviewed or completed a patient questionnaire, or other?

Please be more cautiously in the interpretation in consideration of risk of bias (e.g. self-reported adherence measures, sensible questions) (STROBE item 20). No information is given on generalizability (STROBE item 21).

Further information can be found here: https://journals.plos.org/plosmedicine/article/file?id=10.1371/journal.pmed.0040297&type=printable

We would appreciate receiving your revised manuscript by Feb 02 2020 11:59PM. To enhance the reproducibility of your results, we recommend that if applicable you deposit your laboratory protocols in protocols.io, where a protocol can be assigned its own identifier (DOI) such that it can be cited independently in the future. For instructions see: http://journals.plos.org/plosone/s/submission-guidelines#loc-laboratory-protocols

We look forward to receiving your revised manuscript.

Kind regards,

Tim Mathes

Academic Editor

PLOS ONE

---

## [Author Response · Author response to Decision Letter 2]

23 Dec 2019

23rd Dec 2019

The Editor

PLOS ONE

Dear Sir/Madam,

RE: SUBMISSION OF REVISED MANUSCRIPT

I am submitting a revised manuscript entitled “Psycho-behavioural factors associated with medication adherence and health-related quality of life of male out-patients with hypertension in a Ghanaian hospital” for publication in PLOS ONE journal. All comments have been addressed point-by-point as suggested by the reviewers. There are no legal restrictions on sharing a de-identified data set. The results have been reviewed according to STROBE statement for cross-sectional studies as shown below. 

Comments Response 

There is still no information on missing data in the statistic section (STROBE item 12c) as well as in the results section (STROBE item 13).

 Multiple imputation by chained equation was used to impute for missing information using the predictive mean matching imputation method. Appendix 1 (table 5) shows the questionnaire item response rates (page 5)

In addition, please provide information who performed the assessment. Were the patients interviewed or completed a patient questionnaire, or other Three research assistants were trained for three days for the interviewer-assisted data collection process. The research assistants read the questions to the respondents, and completed the questionnaires based on the respondents’ answers. (Page 3)

Please be more cautiously in the interpretation in consideration of risk of bias (e.g. self-reported adherence measures, sensible questions) (STROBE item 20 these self-reported measures may be prone to recall bias. To reduce the potential of recall bias, we limited the timeline during which these behaviours occurred to the most recent 4-week period. (page 8)

No information is given on generalizability (STROBE item 21).

Further information can be found here We also note that, this was a study conducted in a teaching hospital in Ghana so the findings cannot be generalised to all male patients with hypertension in Ghana.

(page 8)

Data Attached as new compressed zipped folder

All authors declare no conflict of interest. 

I will be grateful if this manuscript can be considered for publication in your journal. 

Thank you. 

Yours Sincerely

Dr. Vincent Boima

---

## [Editor Report · Decision Letter 3]

2 Jan 2020

Psycho-behavioural factors associated with medication adherence among male out-patients with hypertension in a Ghanaian hospital

PONE-D-19-23246R3

Dear Dr. Boima,

We are pleased to inform you that your manuscript has been judged scientifically suitable for publication and will be formally accepted for publication once it complies with all outstanding technical requirements.

With kind regards,

Tim Mathes

Academic Editor

PLOS ONE
---

## [Editor Report · Acceptance letter]

13 Jan 2020

PONE-D-19-23246R3 

Psycho-behavioural factors associated with medication adherence among male out-patients with hypertension in a Ghanaian hospital 

Dear Dr. Boima:

I am pleased to inform you that your manuscript has been deemed suitable for publication in PLOS ONE. Congratulations! Your manuscript is now with our production department. 

With kind regards,

on behalf of

Dr. Tim Mathes 

Academic Editor

PLOS ONE